# Occurrence and transmission potential of asymptomatic and presymptomatic SARS-CoV-2 infections: A living systematic review and meta-analysis

**Diana Buitrago-Garcia**[1,2☯], **Dianne Egli-Gany**[1☯], **Michel J. Counotte**[1☯],
**Stefanie Hossmann**[1], **Hira Imeri**[1], **Aziz Mert Ipekci**[1], **Georgia Salanti**[1],
**Nicola Low**[1]*

**1** Institute of Social and Preventive Medicine, University of Bern, Bern, Switzerland, **2** Graduate School of Health Sciences, University of Bern, Bern, Switzerland

☯ These authors contributed equally to this work.
* nicola.low@ispm.unibe.ch

**Data Availability Statement:** The file listing all included studies and files used for all analyses are available from the Harvard Dataverse database.

## Abstract

### Background

There is disagreement about the level of asymptomatic severe acute respiratory syndrome coronavirus 2 (SARS-CoV-2) infection. We conducted a living systematic review and meta-analysis to address three questions: (1) Amongst people who become infected with SARS-CoV-2, what proportion does not experience symptoms at all during their infection? (2) Amongst people with SARS-CoV-2 infection who are asymptomatic when diagnosed, what proportion will develop symptoms later? (3) What proportion of SARS-CoV-2 transmission is accounted for by people who are either asymptomatic throughout infection or presymptomatic?

### Methods and findings

We searched PubMed, Embase, bioRxiv, and medRxiv using a database of SARS-CoV-2 literature that is updated daily, on 25 March 2020, 20 April 2020, and 10 June 2020. Studies of people with SARS-CoV-2 diagnosed by reverse transcriptase PCR (RT-PCR) that documented follow-up and symptom status at the beginning and end of follow-up or modelling studies were included. One reviewer extracted data and a second verified the extraction, with disagreement resolved by discussion or a third reviewer. Risk of bias in empirical studies was assessed with an adapted checklist for case series, and the relevance and credibility of modelling studies were assessed using a published checklist. We included a total of 94 studies. The overall estimate of the proportion of people who become infected with SARS-CoV-2 and remain asymptomatic throughout infection was 20% (95% confidence interval [CI] 17–25) with a prediction interval of 3%–67% in 79 studies that addressed this review question. There was some evidence that biases in the selection of participants influence the estimate. In seven studies of defined populations screened for SARS-CoV-2 and then

https://doi.org/10.7910/DVN/TZFXYO, Harvard Dataverse, V2, UNF:6:nblmY3m4rXPJ/oD2d9Lo5A== [fileUNF].

**Funding:** Funding was received from the Swiss National Science Foundation (320030_176233, to NL), http://www.snf.ch/en/Pages/default.aspx; European Union Horizon 2020 research and innovation programme (101003688, to NL), https://ec.europa.eu/programmes/horizon2020/en; Swiss government excellence scholarship (2019.0774, to DB-G), https://www.sbfi.admin.ch/sbfi/en/home/education/scholarships-and-grants/swiss-government-excellence-scholarships.html; and the Swiss School of Public Health Global P3HS stipend (to DB-G), https://ssphplus.ch/en/globalp3hs/. The funders had no role in study design, data collection and analysis, decision to publish, or preparation of the manuscript.

**Competing interests:** I have read the journal's policy and the authors of this manuscript have the following competing interests: GS has participated in two scientific meetings for Merck and Biogen. NL is a member of the PLOS Medicine editorial board.

**Abbreviations:** CI, confidence interval; COVID-19, coronavirus disease 2019; CrI, credibility interval; E, number of secondary transmission events; F, female; GI, generation interval; IQR, interquartile range; M, male; NR, not reported; RT-PCR, reverse transcriptase PCR; SARS-CoV-2, severe acute respiratory syndrome coronavirus 2; USA, United States of America.

followed, 31% (95% CI 26%–37%, prediction interval 24%–38%) remained asymptomatic. The proportion of people that is presymptomatic could not be summarised, owing to heterogeneity. The secondary attack rate was lower in contacts of people with asymptomatic infection than those with symptomatic infection (relative risk 0.35, 95% CI 0.10–1.27). Modelling studies fit to data found a higher proportion of all SARS-CoV-2 infections resulting from transmission from presymptomatic individuals than from asymptomatic individuals. Limitations of the review include that most included studies were not designed to estimate the proportion of asymptomatic SARS-CoV-2 infections and were at risk of selection biases; we did not consider the possible impact of false negative RT-PCR results, which would underestimate the proportion of asymptomatic infections; and the database does not include all sources.

## Conclusions

The findings of this living systematic review suggest that most people who become infected with SARS-CoV-2 will not remain asymptomatic throughout the course of the infection. The contribution of presymptomatic and asymptomatic infections to overall SARS-CoV-2 transmission means that combination prevention measures, with enhanced hand hygiene, masks, testing tracing, and isolation strategies and social distancing, will continue to be needed.

---

## Author summary

### Why was this study done?

- The proportion of people who will remain asymptomatic throughout the course of infection with severe acute respiratory syndrome coronavirus 2 (SARS-CoV-2), the cause of coronavirus disease 2019 (COVID-19), is not known.

- Studies that assess people at just one time point will overestimate the proportion of true asymptomatic infection because those who go on to develop COVID-19 symptoms will be wrongly classified as asymptomatic rather than presymptomatic.

- The amount, and infectiousness, of asymptomatic SARS-CoV-2 infection will determine what kind of measures will prevent transmission most effectively.

### What did the researchers do and find?

- We did a living systematic review through 10 June 2020, using automated workflows that speed up the review processes and allow the review to be updated when relevant new evidence becomes available.

- Overall, in 79 studies in a range of different settings, 20% (95% confidence interval [CI] 17%–25%, prediction interval 3%–67%) of people with SARS-CoV-2 infection remained asymptomatic during follow-up, but biases in study designs limit the certainty of this estimate.

- In seven studies of defined populations screened for SARS-CoV-2 and then followed, 31% (95% CI 26%–37%, prediction interval 24%–38%) remained asymptomatic.

- We found some evidence that SARS-CoV-2 infection in contacts of people with asymptomatic infection is less likely than in contacts of people with symptomatic infection (relative risk 0.35, 95% CI 0.10–1.27).

## What do these findings mean?

- The findings of this living systematic review suggest that most people who become infected with SARS-CoV-2 will not remain asymptomatic throughout the course of infection.

- Future studies should be designed specifically to determine the true proportion of asymptomatic SARS-CoV-2 infections, using methods to minimise biases in the selection of study participants and ascertainment of symptom status during follow-up.

- The contribution of presymptomatic and asymptomatic infections to overall SARS-CoV-2 transmission means that combination prevention measures, with enhanced hand hygiene, masks, testing tracing, and isolation strategies and social distancing, will continue to be needed.

## Introduction

There is ongoing discussion about the level of asymptomatic severe acute respiratory syndrome coronavirus 2 (SARS-CoV-2) infection. The authors of a narrative review report a range of proportions of participants positive for SARS-CoV-2 but asymptomatic in different studies from 6% to 96% [1]. The discrepancy results, in part, from the interpretation of studies that report a proportion of asymptomatic people with SARS-CoV-2 detected at a single point. The studies cited include both people who will remain asymptomatic throughout and those, known as presymptomatic, who will develop symptoms of coronavirus disease 2019 (COVID-19) if followed up [2]. The full spectrum and distribution of COVID-19, from completely asymptomatic, to mild and nonspecific symptoms, viral pneumonia, respiratory distress syndrome, and death, are not yet known [3]. Without follow-up, however, the proportions of asymptomatic and presymptomatic infections cannot be determined.

Accurate estimates of the proportions of true asymptomatic and presymptomatic infections are needed urgently because their contribution to overall SARS-CoV-2 transmission at the population level will determine the appropriate balance of control measures [3]. If the predominant route of transmission is from people who have symptoms, then strategies should focus on testing, followed by isolation of infected individuals and quarantine of their contacts. If, however, most transmission is from people without symptoms, social distancing measures that reduce contact with people who might be infectious should be prioritised, enhanced by active case-finding through testing of asymptomatic people.

The objectives of this study were to address three questions: (1) Amongst people who become infected with SARS-CoV-2, what proportion do not experience symptoms at all during their infection? (2) Amongst people with SARS-CoV-2 infection who are asymptomatic when diagnosed, what proportion will develop symptoms later? (3) What proportion of SARS-CoV-2 transmission is accounted for by people who are either asymptomatic throughout infection or presymptomatic?

## Methods

We conducted a living systematic review, a systematic review that provides an online summary of findings and is updated when relevant new evidence becomes available [4]. The review follows a published protocol (https://osf.io/9ewys/), which describes in detail the methods used to speed up review tasks [5] and to assess relevant evidence rapidly during a public health emergency [6]. The first two versions of the review have been published as preprints [7,8]. We report our findings according to the statement on preferred reporting items for systematic reviews and meta-analyses (S1 PRISMA Checklist) [9]. Ethics committee review was not required for this study. Box 1 shows our definitions of symptoms, asymptomatic infection, and presymptomatic status. We use the term asymptomatic SARS-CoV-2 infection for people without symptoms of COVID-19 who remain asymptomatic throughout the course of infection. We use the term presymptomatic for people who do not have symptoms of COVID-19 when enrolled in a study but who develop symptoms during adequate follow-up.

---

### Box 1. Definitions of symptoms and symptom status in a person with SARS-CoV-2 infections

**Symptoms:** symptoms that a person experiences and reports. We used the authors' definitions. We searched included manuscripts for an explicit statement that the study participant did not report symptoms that they experienced. Some authors defined 'asymptomatic' as an absence of self-reported symptoms. We did not include clinical signs observed or elicited on examination.

**Asymptomatic infection:** a person with laboratory-confirmed SARS-CoV-2 infection, who has no symptoms, according to the authors' report, at the time of first clinical assessment and had no symptoms at the end of follow-up. The end of follow-up was defined as any of the following: virological cure, with one or more negative reverse transcriptase PCR (RT-PCR) test results; follow-up for 14 days or more after the last possible exposure to an index case; follow-up for 7 days or more after the first RT-PCR positive result.

**Presymptomatic:** a person with laboratory-confirmed SARS-CoV-2 infection, who has no symptoms, according to the authors' report, at the time of first clinical assessment but who developed symptoms by the end of follow-up. The end of follow-up was defined as any of the following: virological cure, with one or more negative RT-PCR test results; follow-up for 14 days or more after the last possible exposure to an index case; follow-up for 7 days or more after the first RT-PCR positive result.

---

### Information sources and search

We conducted the first search on 25 March 2020 and updated it on 20 April and 10 June 2020. We searched the COVID-19 living evidence database [10], which is generated using automated workflow processes [5] to (1) provide daily updates of searches of four electronic databases (Medline PubMed, Ovid Embase, bioRxiv, and medRxiv), using medical subject headings and free-text keywords for SARS-CoV-2 infection and COVID-19; (2) de-duplicate the records; (3) tag records that are preprints; and (4) allow searches of titles and abstracts using Boolean operators. We used the search function to identify studies of asymptomatic or presymptomatic SARS-CoV-2 infection using a search string of medical subject headings and free-text

keywords (S1 Text). We also examined articles suggested by experts and the reference lists of retrieved mathematical modelling studies and systematic reviews. Reports from this living rapid systematic review will be updated at 3-monthly intervals, with continuously updated searches.

## Eligibility criteria

We included studies in any language of people with SARS-CoV-2 diagnosed by RT-PCR that documented follow-up and symptom status at the beginning and end of follow-up or investigated the contribution to SARS-CoV-2 transmission of asymptomatic or presymptomatic infection. We included contact-tracing investigations, case series, cohort studies, case-control studies, and statistical and mathematical modelling studies. We excluded the following study types: case reports of a single patient and case series in which participants were not enrolled consecutively. When multiple records included data from the same study population, we linked the records and extracted data from the most complete report.

## Study selection and data extraction

Reviewers worked in pairs to screen records using an application programming interface in the electronic data capture system (REDCap, Vanderbilt University, Nashville, TN, USA). One reviewer selected potentially eligible studies and a second reviewer verified all included and excluded studies. We reported the identification, exclusion, and inclusion of studies in a flowchart (S1 Fig). The reviewers determined which of the three review questions each study addressed, using the definitions in Box 1. One reviewer extracted data using a pre-piloted extraction form in REDCap, and a second reviewer verified the extracted data using the query system. A third reviewer adjudicated on disagreements that could not be resolved by discussion. We contacted study authors for clarification when the study description was insufficient to reach a decision on inclusion or if reported data in the manuscript were internally inconsistent. The extracted variables included, but were not limited to, study design, country and/or region, study setting, population, age, primary outcomes, and length of follow-up. From empirical studies, we extracted raw numbers of individuals with any outcome and its relevant denominator. From statistical and mathematical modelling studies, we extracted proportions and uncertainty intervals reported by the authors.

The primary outcomes for each review question were (1) proportion with asymptomatic SARS-CoV-2 infection who did not experience symptoms at all during follow-up; (2) proportion with SARS-CoV-2 infections who did not have symptoms at the time of testing but developed symptoms during follow-up; (3) estimated proportion (with uncertainty interval) of SARS-CoV-2 transmission accounted for by people who are asymptomatic or presymptomatic. A secondary outcome for review question 3 was the secondary attack rate from asymptomatic or presymptomatic index cases.

## Risk of bias in included studies

Two authors independently assessed the risk of bias. A third reviewer resolved disagreements. For observational epidemiological studies, we adapted the Joanna Briggs Institute Critical Appraisal Checklist for Case Series [11]. The adapted tool included items about inclusion criteria, measurement of asymptomatic status, follow-up of course of disease, and statistical analysis. We added items about selection biases affecting the study population from a tool for the assessment of risk of bias in prevalence studies [12]. For mathematical modelling studies, we used a checklist for assessing relevance and credibility [13].

## Synthesis of the evidence

We used the 'metaprop' and 'metabin' functions from the 'meta' package (version 4.11–0) [14] in R (version 3.5.1) to display the study findings in forest plots and synthesise their findings. The 95% confidence intervals (CIs) for each study are estimated using the Clopper-Pearson method [15]. We examined heterogeneity visually in forest plots. We stratified studies according to the methods used to identify people with asymptomatic SARS-CoV-2 infection and the study setting. To synthesise proportions from comparable studies, in terms of design and population, we used stratified random-effects meta-analysis. For the stratified and overall summary estimates, we calculated prediction intervals, to represent the likely range of proportions that would be obtained in subsequent studies conducted in similar settings [16]. We calculated the secondary attack rate as the number of cases among contacts as a proportion of all close contacts ascertained. We did not account for potential clustering of contacts because the included studies did not report the size of clusters. We compared the secondary attack rate from asymptomatic or presymptomatic index cases with that from symptomatic cases. If there were no events in a group, we added 0.5 to each cell in the 2 × 2 table. We used random-effects meta-analysis with the Mantel-Haenszel method to estimate a summary risk ratio (with 95% CI).

## Results

The living evidence database contained a total of 25,538 records about SARS-CoV-2 or COVID-19 by 10 June 2020. The searches for studies about asymptomatic or presymptomatic SARS-CoV-2 on 25 March, 20 April, and 10 June resulted in 89, 230, and 688 records for screening (S1 Fig). In the first version of the review [7], 11 articles were eligible for inclusion [17–27], version 2 [8] identified another 26 eligible records [28–53], and version 3 identified another 61 eligible records [54–114]. After excluding four articles for which more recent data became available in a subsequent version [25,29,30,35], the total number of articles included was 94 (S1 Table) [17–24,26–28,31–34,36–114]. The types of evidence changed across the three versions of the review (S1 Table). In the first version, six of 11 studies were contact investigations of single-family clusters with a total of 39 people. In the next versions, study designs included larger investigations of contacts and outbreaks, screening of defined groups, and studies of hospitalised adults and children. Across all three review versions, data from 79 empirical observational studies were collected in 19 countries or territories (Tables 1 and 2) and included 6,832 people with SARS-CoV-2 infection. Forty-seven of the studies, including 3,802 infected people, were done in China (S2 Table). At the time of their inclusion in the review, 23 of the included records were preprints; six of these had been published in peer-reviewed journals by 17 July 2020 [19,20,27,81,82,106].

### Proportion of people with asymptomatic SARS-CoV-2 infection

We included 79 studies that reported empirical data about 6,616 people with SARS-CoV-2 infection (1,287 defined as having asymptomatic infection) [17,18,21–23,26–28,31,32,34,36, 39–45,47–50,52–54,56–62,64,66–68,70–77,79–90,92–112,114] and one statistical modelling study [24] (Table 1). The sex distribution of the people with asymptomatic infection was reported in 41/79 studies, and the median age was reported in 35/79 studies (Table 1). The results of the studies were heterogeneous (S2 Fig). We defined seven strata, according to the method of selection of asymptomatic status and study settings. Study findings within some of these strata were more consistent (Fig 1). We considered the statistical modelling study of passengers on the *Diamond Princess* cruise ship passengers [24] separately, because of the different

**Table 1. Characteristics of studies reporting on proportion of asymptomatic SARS-CoV-2 infections.**

| Author | Country, location | Total SARS-CoV-2, n | Asymptomatic SARS-CoV-2, n | Sex of asymptomatic people | Age of asymptomatic people, years, median | Follow-up method[a] |
|---|---|---|---|---|---|---|
| Contact investigation, single | | | | | | |
| Tong, ZD [44] | China, Zhejiang | 5 | 3 | 2 F, 3 M | 28 IQR 12–41 | 1, 3 |
| Huang, R [74] | China, Suqian | 2 | 1 | 1 F, 0 M | 54 | 3 |
| Jiang, XL [76] | China, Shandong | 8 | 3 | 3 F, 0 M | 35 IQR 0–53 | 3 |
| Jiang, X [75] | China, Chongqing | 3 | 1 | 1 F, 0 M | 8 | 2 |
| Liao, J [22] | China, Chongqing | 12 | 3 | NR | NR | 1, 2 |
| Hu, Z [21] | China, Nanjing | 4 | 1 | 0 F, 1 M | 64 | 2, 3 |
| Luo, SH [23] | China, Anhui | 4 | 1 | 1 F, 0 M | 50 | 1, 2, 3 |
| Chan, JF [18] | China, Guangdong | 5 | 1 | 0 F, 1 M | 10 | 1 |
| Ye, F [49] | China, Sichuan | 5 | 1 | 0 F, 1 M | 28 | 1, 2 |
| Bai, Y [17] | China, Anyang | 6 | 1 | 1 F, 0 M | 20 | 1 |
| Luo, Y [85] | China, Wuhan | 6 | 5 | NR | 37 IQR 7–62 | 1 |
| Zhang, J [50] | China, Wuhan and Beijing | 5 | 2 | 1 F, 1 M | NR | 2 |
| Zhang, B [110] | China, Guangdong | 7 | 2 | 0 F, 2 M | 13.5 IQR 13–14 | 3 |
| Huang, L [73] | China, Gansu | 7 | 2 | 2 F, 0 M | 44 IQR 38.5–49.5 | 2 |
| Qian, G [26] | China, Zhejiang | 8 | 2 | 1 F, 1 M | 30.5 IQR 1–60 | 1, 2 |
| Gao, Y [70] | China, Wuxi | 15 | 6 | 3 F, 3 M | 50 IQR 48–51 | 1, 2 |
| Contact investigation, aggregated | | | | | | |
| Hijnen, D [72] | Germany | 11 | 1 | 0 F, 1 M | 49 | 1 |
| Brandstetter, S [62] | Germany | 36 | 2 | NR | NR | 2 |
| Zhang, W2 [111] | China, Guiyang | 12 | 4 | NR | NR | 1, 2, 3 |
| Cheng, HY [66] | Taiwan | 22 | 4 | NR | NR | 1 |
| Wang, Z [47] | China, Wuhan | 47 | 4 | NR | NR | 1 |
| Wu, J [105] | China, Zhuhai | 83 | 8 | NR | NR | 1, 2 |
| Luo, L [36] | China, Guangzhou | 129 | 8 | NR | NR | 1, 2, 3 |
| Bi, Q [60] | China, Shenzhen | 87 | 17 | NR | NR | 2, 3 |
| Yang, R [108] | China, Wuhan | 78 | 33 | 22 F, 11 M | 37 IQR 26–45 | 3 |
| Outbreak investigation | | | | | | |
| Danis, K [32] | France | 13 | 1 | NR | NR | 1, 2 |
| Böhmer, MM [61] | Germany | 16 | 1 | NR | NR | 1 |
| Roxby, AC [94] | USA | 6 | 3 | NR | NR | 1 |
| Yang, N [48] | China, Xiaoshan | 10 | 2 | 1 F, 1 M | NR | 1, 2 |
| Schwierzeck, V [95] | Germany | 12 | 2 | NR | NR | 2 |
| Arons, MM [58] | USA | 47 | 3 | NR | NR | 2 |
| Park, SY [90] | South Korea | 97 | 4 | NR | NR | 2 |
| Dora, AV [68] | USA | 19 | 6 | 0 F, 6 M | 75 IQR 72–75 | 3 |
| Tian, S [43] | China, Shandong | 24 | 7 | NR | NR | 3 |
| Solbach, W [97] | Germany | 97 | 10 | NR | NR | 2 |
| Graham, N [71] | United Kingdom | 126 | 46 | NR | NR | 2 |

*(Continued)*

**Table 1.** (Continued)

| Author | Country, location | Total SARS-CoV-2, n | Asymptomatic SARS-CoV-2, n | Sex of asymptomatic people | Age of asymptomatic people, years, median | Follow-up method[a] |
|---|---|---|---|---|---|---|
| Pham, TQ [100] | Vietnam | 208 | 89 | NR | 31 IQR 23–45 | 2 |
| Screening of defined population | | | | | | |
| Hoehl, S [34] | Germany | 2 | 1 | 0 F, 1 M | 58 | 2 |
| Chang, L [31] | China, Wuhan | 4 | 2 | 0 F, 2 M | 45 IQR 37–53 | 2 |
| Arima, Y [28] | Japan | 12 | 4 | NR | NR | 1, 2 |
| Rivett, L [93] | United Kingdom | 30 | 5 | NR | NR | 2 |
| Treibel, TA [101] | United Kingdom | 44 | 12 | NR | NR | 2 |
| Lavezzo, E [81] | Italy | 73 | 29 | NR | NR | 2 |
| Lombardi, A [82] | Italy | 138 | 41 | NR | NR | 3 |
| Hospitalised adults | | | | | | |
| Pongpirul, WA [39] | Thailand | 11 | 1 | 1 F, 0 M | 66 | 2, 3 |
| Zou, L [53] | China, Zhuhai | 18 | 1 | 1 M, 0 M | 26 | 1 |
| Qiu, C [92] | China, Hunan | 104 | 5 | NR | NR | 2 |
| Zhou, R [114] | China, Guangdong | 31 | 9 | NR | NR | 3 |
| Chang, MC [64] | South Korea | 139 | 10 | 4 F, 6 M | NR | 1, 2 |
| Zhou, X [52] | China, Shanghai | 328 | 10 | NR | NR | 1, 2, 3 |
| Angelo Vaira, L [57] | Italy | 345 | 10 | NR | NR | 3 |
| Wang, X [45] | China, Wuhan | 1012 | 14 | NR | NR | 1, 2 |
| Wong, J [103] | Brunei | 138 | 16 | NR | NR | 2, 3 |
| Xu, T [107] | China, Jiangsu | 342 | 15 | 5 F, 10 M | 27 IQR 17–36 | 2, 3 |
| London, V [83] | USA | 68 | 22 | 22 F, 0 M | 30.5 IQR 24.5–34.8 | 2 |
| Tabata, S [27] | Japan[b] | 104 | 33 | 18 F, 15 M | 70 IQR 57–75 | 2 |
| Andrikopoulou, M [56] | USA | 158 | 46 | 46 F, 0 M | NR | 1, 2 |
| Noh, JY [89] | South Korea | 199 | 53 | NR | NR | 3 |
| Kumar, R [80] | India, New Delhi | 231 | 108 | 18 F, 90 M | NR | 2, 3 |
| Hospitalised children | | | | | | |
| See, KC [41] | Malaysia | 4 | 1 | 0 F, 1 M | 9 | 1, 2, 3 |
| Tan, YP [42] | China, Changsha | 10 | 2 | 1 F, 1 M | 8 | 2, 3 |
| Tan, X [99] | China, Changsha | 13 | 2 | 2 F, 0 M | 5 IQR 2–8 | 1, 2, 3 |
| Melgosa, M [87] | Spain | 16 | 3 | NR | NR | 1, 2 |
| Wu, HP [104] | China, Jiangxi | 23 | 3 | NR | NR | 3 |
| Song, W [98] | China, Hubei | 16 | 8 | 3 F, 5 M | 11 IQR 7–12 | 1, 2 |
| Bai, K [59] | China, Chongqing | 25 | 8 | NR | NR | 3 |
| Xu, H [106] | China, Guizhou | 32 | 11 | 4 F, 7 M | NR | 1, 2 |
| Qiu, H [40] | China, Zhejiang | 36 | 10 | NR | NR | 1, 2, 3 |
| Lu, Y [84] | China, Wuhan | 110 | 29 | 12 F, 17 M | 7 IQR 6–11 | 2, 3 |
| Hospitalised adults and children | | | | | | |
| Merza, MA [88] | Iraqi Kurdistan | 15 | 6 | NR | NR | 2, 3 |
| Yongchen, Z [109] | China, Jiangsu | 21 | 5 | 2 F, 3 M | 25 IQR 14–54 | 1, 2, 3 |

*(Continued)*

**Table 1.** (Continued)

| Author | Country, location | Total SARS-CoV-2, *n* | Asymptomatic SARS-CoV-2, *n* | Sex of asymptomatic people | Age of asymptomatic people, years, median | Follow-up method[a] |
|---|---|---|---|---|---|---|
| Ma, Y [86] | China, Shandong | 47 | 11 | 5 F, 6 M | 23 IQR NR | 2 |
| Kim, SE [77] | South Korea | 71 | 10 | 6 F, 4 M | 31 IQR 21–55 | 2 |
| Choe, PG [67] | South Korea | 113 | 15 | 17 F, 8 M | NR | 3 |
| Sharma, AK [96] | India, Jaipur | 234 | 215 | NR | NR | 1, 2, 3 |
| Zhang, W3 [112] | China, Guiyang | 137 | 26 | 12 F, 14 M | 24 IQR 12–36 | 1, 2 |
| Alshami, AA [54] | Saudi Arabia | 128 | 69 | 36 F, 33 M | NR | 2, 3 |
| Kong, W [79] | China, Sichuan | 473 | 45 | NR | NR | 1, 2 |
| Wang, Y2 [102] | China, Chongqing | 279 | 63 | 29 F, 34 M | 39 IQR 27–53 | 3 |

[a]Follow-up according to protocol (1: 14 days after last possible exposure; 2: 7 days after diagnosis; 3: until negative RT-PCR result).

[b]People of different nationalities taken from *Diamond Princess* cruise ship to a hospital in Japan.

Abbreviations: F, female; IQR, interquartile range; M, male; NR, not reported; RT-PCR, reverse transcriptase PCR; SARS-CoV-2, severe acute respiratory syndrome coronavirus 2; USA, United States of America

method of analysis and overlap with the study population reported by Tabata and colleagues [27].

The main risks of bias across all categories of empirical studies were in the selection and enrolment of people with asymptomatic infection and mismeasurement of asymptomatic status because of absent or incomplete definitions (S3 Fig). Sources of bias specific to studies in particular settings are discussed with the relevant results.

The overall estimate of the proportion of people who become infected with SARS-CoV-2 and remain asymptomatic throughout the course of infection was 20% (95% CI 17%–25%, 79 studies), with a prediction interval of 3%–67% (Fig 1). One statistical modelling study was based on data from all 634 passengers from the *Diamond Princess* cruise ship with RT-PCR positive test results [24]. The authors adjusted for the proportion of people who would develop symptoms (right censoring) in a Bayesian framework to estimate that, if all were followed up until the end of the incubation period, the probability of asymptomatic infections would be 17.9% (95% credibility interval [CrI] 15.5%–20.2%).

The summary estimates of the proportion of people with asymptomatic SARS-CoV-2 infection differed according to study setting, although prediction intervals for all groups overlapped. The first three strata in Fig 1 involve studies that reported on different types of contact investigation, which start with an identified COVID-19 case. The studies reporting on single-family clusters (21 estimates from 16 studies in China, *n* = 102 people with SARS-CoV-2) all included at least one asymptomatic person [17,18,21–23,26,44,49,50,70,73–76,85,110]. The summary estimate was 34% (95% CI 26%–44%, prediction interval 25%–45%). In nine studies that reported on close contacts of infected individuals and aggregated data from clusters of both asymptomatic and symptomatic people with SARS-CoV-2 the summary estimate was 14% (95% CI 8%–23%, prediction interval 2%–53%) [36,47,60,62,66,72,105,108,111]. We included 12 studies (*n* = 675 people) that reported on outbreak investigations arising from a single symptomatic person or from the country's first imported cases of people with COVID-19 [32,43,48,58,61,68,71,90,94,95,97,100]. Four of the outbreaks involved nursing homes [58,68,71,94] and four involved occupational settings [43,61,90,95]. The summary estimate of

**Table 2. Characteristics of studies that measured the proportion of people with SARS-CoV-2 infection that develops symptoms.**

| Author | Country, location | Total asymptomatic SARS-CoV-2, n | Develop symptoms after testing, n | Sex of asymptomatic people at time of testing | Age of asymptomatic people at time of testing, years, median | Follow-up method[a] |
|---|---|---|---|---|---|---|
| Contact investigation, single | | | | | | |
| Ye, F [49] | China, Sichuan | 3 | 2 | 0 F, 3 M | 28 IQR 23–50 | 1, 2 |
| Zhang, B [110] | China, Guangdong | 4 | 2 | 0 F, 4 M | 34 IQR 33–35 | 3 |
| Huang, L [73] | China, Gansu | 4 | 2 | 3 F, 1 M | 44.5 IQR 34.50–54.25 | 2 |
| Jiang, XL [76] | China, Shandong | 5 | 2 | 3 F, 2 M | 35 IQR 35–37 | 3 |
| Hu, Z [21] | China, Nanjing | 24 | 5 | NR | NR | 2, 3 |
| Contact investigation, aggregated | | | | | | |
| Zhang, W2 [111] | China, Guangzhou | 12 | 8 | NR | NR | 1, 2, 3 |
| Outbreak investigation | | | | | | |
| Schwierzeck, V [95] | Germany | 6 | 4 | NR | NR | 2 |
| Park, SY [90] | South Korea | 8 | 4 | NR | NR | 2 |
| Arons, MM [58] | USA | 27 | 24 | NR | NR | 2 |
| Dora, AV [68] | USA | 14 | 8 | 0 F, 14 M | NR | 3 |
| Graham, N [71] | United Kingdom | 54 | 8 | NR | NR | 1 |
| Screening of defined population | | | | | | |
| Hoehl, S [34] | Germany | 2 | 1 | 1 F, 1 M | 51 | 2 |
| Rivett, L [93] | United Kingdom | 6 | 1 | NR | NR | 2 |
| Chang, L [31] | China, Wuhan | 4 | 2 | 1 F, 3 M | 39.5 IQR 29–47.5 | 2 |
| Arima, Y [28] | Japan | 5 | 2 | NR | NR | 1, 2 |
| Lytras, T [37] | Greece | 39 | 4 | NR | NR | 2 |
| Lavezzo, E [81] | Italy | 39 | 10 | NR | NR | 2 |
| Hospitalised adults | | | | | | |
| Al-Shamsi, HO [55] | United Arab Emirates | 7 | 7 | 5 F, 2 M | 51.6 IQR 40–76 | 3 |
| Luo, SH [23] | China, Anhui | 8 | 7 | NR | NR | 1, 2, 3 |
| Zhou, X [52] | China, Shanghai | 13 | 3 | 7 F, 6 M | NR | 2, 3 |
| Zhou, R [114] | China, Guangdong | 31 | 22 | NR | NR | 3 |
| Wang, X [45] | China, Wuhan | 30 | 16 | NR | NR | 1, 2 |
| Tabata, S [27] | Cruise Ship | 43 | 10 | 24 F, 19 M | 69 IQR 60.5–75 | 2 |
| Wang, Y1 [46] | China, Shenzhen | 55 | 43 | NR | 49 IQR 2–69 | 3 |
| Meng, H [38] | China, Wuhan | 58 | 16 | NR | NR | 2 |
| Andrikopoulou, M [56] | USA | 63 | 16 | 63 F, 0 M | NR | 1, 2 |
| Zhang, Z [113] | China, Shenzhen | 56 | 33 | 33 F, 23 M | NR | 2, 3 |
| Wong, J [103] | Brunei | 138 | 42 | NR | NR | 2, 3 |
| Hospitalised children | | | | | | |

(Continued)

**Table 2.** (Continued)

| Author | Country, location | Total asymptomatic SARS-CoV-2, *n* | Develop symptoms after testing, *n* | Sex of asymptomatic people at time of testing | Age of asymptomatic people at time of testing, years, median | Follow-up method[a] |
|---|---|---|---|---|---|---|
| See, KC [41] | Malaysia | 2 | 1 | 0 F, 2 M | 5 IQR 1–9 | 1, 2, 3 |
| Hospitalised adults and children | | | | | | |
| Kim, SE [77] | South Korea | 13 | 3 | 7 F, 6 M | 31 IQR 20.5–51.5 | 2 |
| Choe, PG [67] | South Korea | 54 | 39 | 32 F, 22 M | NR | 3 |
| Kong, W [79] | China, Sichuan | 62 | 17 | NR | NR | 1 |

[a]Follow-up according to protocol (1: 14 days after possible exposure; 2: 7 days after diagnosis; 3: until one or more negative RT-PCR result).

[b]People of different nationalities taken from *Diamond Princess* cruise ship to a hospital in Japan.

[c]Until hospital discharge or negative RT-PCR.

Abbreviations: F, female; IQR, interquartile range; M, male; NR, not reported; RT-PCR, reverse transcriptase PCR; SARS-CoV-2, severe acute respiratory syndrome coronavirus 2; USA, United States of America

the proportion of asymptomatic SARS-CoV-2 infections was 18% (95% CI 10%–28%, prediction interval 2%–64%).

In seven studies, people with SARS-CoV-2 infection were detected through screening of all people in defined populations who were potentially exposed (303 infected people amongst 10,090 screened) [28,31,34,81,82,93,101]. The screened populations included healthcare workers [82,93,101]; people evacuated from a setting where SARS-CoV-2 transmission was confirmed, irrespective of symptom status [28,34]; the whole population of one village in Italy [81]; and blood donors [31]. In these studies, the summary estimate of the proportion asymptomatic was 31% (95% CI 26%–37%, prediction interval 24%–38%). There is a risk of selection bias in studies of certain groups, such as healthcare workers and blood donors, because people with symptoms are excluded [31,82,93,101], or from nonresponders in population-based screening [81]. Retrospective symptom ascertainment could also increase the proportion determined asymptomatic [81,82,101].

The remaining studies, in hospital settings, included adult patients only (15 studies, *n* = 3,228) [27,39,45,52,53,56,57,64,80,83,89,92,103,107,114], children only (10 studies, *n* = 285) [40–42,59,84,87,98,99,104,106], or adults and children (10 studies, *n* = 1,518) [54,67,77,79,86,88,96,102,109,112] (Table 1, Fig 1). The types of hospital and clinical severity of patients differed, including settings in which anyone with SARS-CoV-2 infection was admitted for isolation and traditional hospitals.

## Proportion of presymptomatic SARS-CoV-2 infections

We included 31 studies in which the people with no symptoms of COVID-19 at enrolment were followed up, and the proportion that develops symptoms is defined as presymptomatic (Table 2, Fig 2) [21,27,28,31,34,37,38,41,45,46,49,52,55,56,58,67,68,71,73,76,77,79,81,90,93, 95,103,110,111,113,114]. Four studies addressed only this review question [37,38,55,113]. The findings from the 31 studies were heterogeneous (S4 Fig), even when categorised according to the method of selection of asymptomatic participants, and we did not estimate a summary measure (Fig 2).

## Additional analyses

We investigated heterogeneity in the estimates of the proportion of asymptomatic SARS-CoV-2 infections in subgroup analyses that were not specified in the original protocol. In studies of

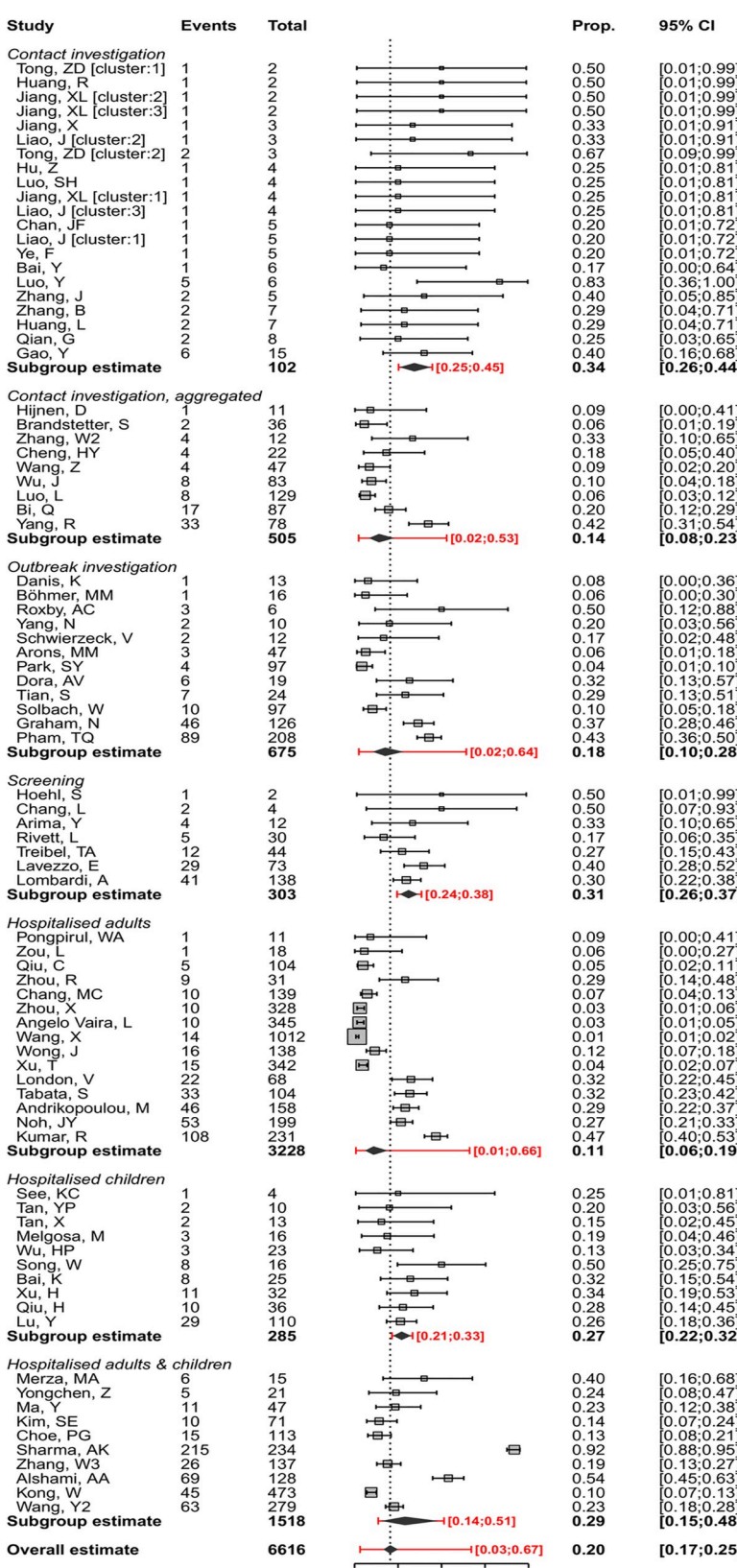

| Study | Events | Total | | Prop. | 95% CI |
|---|---|---|---|---|---|
| *Contact investigation* | | | | | |
| Tong, ZD [cluster:1] | 1 | 2 | | 0.50 | [0.01;0.99] |
| Huang, R | 1 | 2 | | 0.50 | [0.01;0.99] |
| Jiang, XL [cluster:2] | 1 | 2 | | 0.50 | [0.01;0.99] |
| Jiang, XL [cluster:3] | 1 | 2 | | 0.50 | [0.01;0.99] |
| Jiang, X | 1 | 3 | | 0.33 | [0.01;0.91] |
| Liao, J [cluster:2] | 1 | 3 | | 0.33 | [0.01;0.91] |
| Tong, ZD [cluster:2] | 2 | 3 | | 0.67 | [0.09;0.99] |
| Hu, Z | 1 | 4 | | 0.25 | [0.01;0.81] |
| Luo, SH | 1 | 4 | | 0.25 | [0.01;0.81] |
| Jiang, XL [cluster:1] | 1 | 4 | | 0.25 | [0.01;0.81] |
| Liao, J [cluster:3] | 1 | 4 | | 0.25 | [0.01;0.81] |
| Chan, JF | 1 | 5 | | 0.20 | [0.01;0.72] |
| Liao, J [cluster:1] | 1 | 5 | | 0.20 | [0.01;0.72] |
| Ye, F | 1 | 5 | | 0.20 | [0.01;0.72] |
| Bai, Y | 1 | 6 | | 0.17 | [0.00;0.64] |
| Luo, Y | 5 | 6 | | 0.83 | [0.36;1.00] |
| Zhang, J | 2 | 5 | | 0.40 | [0.05;0.85] |
| Zhang, B | 2 | 7 | | 0.29 | [0.04;0.71] |
| Huang, L | 2 | 7 | | 0.29 | [0.04;0.71] |
| Qian, G | 2 | 8 | | 0.25 | [0.03;0.65] |
| Gao, Y | 6 | 15 | | 0.40 | [0.16;0.68] |
| **Subgroup estimate** | | **102** | [0.25;0.45] | **0.34** | **[0.26;0.44]** |
| *Contact investigation, aggregated* | | | | | |
| Hijnen, D | 1 | 11 | | 0.09 | [0.00;0.41] |
| Brandstetter, S | 2 | 36 | | 0.06 | [0.01;0.19] |
| Zhang, W2 | 4 | 12 | | 0.33 | [0.10;0.65] |
| Cheng, HY | 4 | 22 | | 0.18 | [0.05;0.40] |
| Wang, Z | 4 | 47 | | 0.09 | [0.02;0.20] |
| Wu, J | 8 | 83 | | 0.10 | [0.04;0.18] |
| Luo, L | 8 | 129 | | 0.06 | [0.03;0.12] |
| Bi, Q | 17 | 87 | | 0.20 | [0.12;0.29] |
| Yang, R | 33 | 78 | | 0.42 | [0.31;0.54] |
| **Subgroup estimate** | | **505** | [0.02;0.53] | **0.14** | **[0.08;0.23]** |
| *Outbreak investigation* | | | | | |
| Danis, K | 1 | 13 | | 0.08 | [0.00;0.36] |
| Böhmer, MM | 1 | 16 | | 0.06 | [0.00;0.30] |
| Roxby, AC | 3 | 6 | | 0.50 | [0.12;0.88] |
| Yang, N | 2 | 10 | | 0.20 | [0.03;0.56] |
| Schwierzeck, V | 2 | 12 | | 0.17 | [0.02;0.48] |
| Arons, MM | 3 | 47 | | 0.06 | [0.01;0.18] |
| Park, SY | 4 | 97 | | 0.04 | [0.01;0.10] |
| Dora, AV | 6 | 19 | | 0.32 | [0.13;0.57] |
| Tian, S | 7 | 24 | | 0.29 | [0.13;0.51] |
| Solbach, W | 10 | 97 | | 0.10 | [0.05;0.18] |
| Graham, N | 46 | 126 | | 0.37 | [0.28;0.46] |
| Pham, TQ | 89 | 208 | | 0.43 | [0.36;0.50] |
| **Subgroup estimate** | | **675** | [0.02;0.64] | **0.18** | **[0.10;0.28]** |
| *Screening* | | | | | |
| Hoehl, S | 1 | 2 | | 0.50 | [0.01;0.99] |
| Chang, L | 2 | 4 | | 0.50 | [0.07;0.93] |
| Arima, Y | 4 | 12 | | 0.33 | [0.10;0.65] |
| Rivett, L | 5 | 30 | | 0.17 | [0.06;0.35] |
| Treibel, TA | 12 | 44 | | 0.27 | [0.15;0.43] |
| Lavezzo, E | 29 | 73 | | 0.40 | [0.28;0.52] |
| Lombardi, A | 41 | 138 | | 0.30 | [0.22;0.38] |
| **Subgroup estimate** | | **303** | [0.24;0.38] | **0.31** | **[0.26;0.37]** |
| *Hospitalised adults* | | | | | |
| Pongpirul, WA | 1 | 11 | | 0.09 | [0.00;0.41] |
| Zou, L | 1 | 18 | | 0.06 | [0.00;0.27] |
| Qiu, C | 5 | 104 | | 0.05 | [0.02;0.11] |
| Zhou, R | 9 | 31 | | 0.29 | [0.14;0.48] |
| Chang, MC | 10 | 139 | | 0.07 | [0.04;0.13] |
| Zhou, X | 10 | 328 | | 0.03 | [0.01;0.06] |
| Angelo Vaira, L | 10 | 345 | | 0.03 | [0.01;0.05] |
| Wang, X | 14 | 1012 | | 0.01 | [0.01;0.02] |
| Wong, J | 16 | 138 | | 0.12 | [0.07;0.18] |
| Xu, T | 15 | 342 | | 0.04 | [0.02;0.07] |
| London, V | 22 | 68 | | 0.32 | [0.22;0.45] |
| Tabata, S | 33 | 104 | | 0.32 | [0.23;0.42] |
| Andrikopoulou, M | 46 | 158 | | 0.29 | [0.22;0.37] |
| Noh, JY | 53 | 199 | | 0.27 | [0.21;0.33] |
| Kumar, R | 108 | 231 | | 0.47 | [0.40;0.53] |
| **Subgroup estimate** | | **3228** | [0.01;0.66] | **0.11** | **[0.06;0.19]** |
| *Hospitalised children* | | | | | |
| See, KC | 1 | 4 | | 0.25 | [0.01;0.81] |
| Tan, YP | 2 | 10 | | 0.20 | [0.03;0.56] |
| Tan, X | 2 | 13 | | 0.15 | [0.02;0.45] |
| Melgosa, M | 3 | 16 | | 0.19 | [0.04;0.46] |
| Wu, HP | 3 | 23 | | 0.13 | [0.03;0.34] |
| Song, W | 8 | 16 | | 0.50 | [0.25;0.75] |
| Bai, K | 8 | 25 | | 0.32 | [0.15;0.54] |
| Xu, H | 11 | 32 | | 0.34 | [0.19;0.53] |
| Qiu, H | 10 | 36 | | 0.28 | [0.14;0.45] |
| Lu, Y | 29 | 110 | | 0.26 | [0.18;0.36] |
| **Subgroup estimate** | | **285** | [0.21;0.33] | **0.27** | **[0.22;0.32]** |
| *Hospitalised adults & children* | | | | | |
| Merza, MA | 6 | 15 | | 0.40 | [0.16;0.68] |
| Yongchen, Z | 5 | 21 | | 0.24 | [0.08;0.47] |
| Ma, Y | 11 | 47 | | 0.23 | [0.12;0.38] |
| Kim, SE | 10 | 71 | | 0.14 | [0.07;0.24] |
| Choe, PG | 15 | 113 | | 0.13 | [0.08;0.21] |
| Sharma, AK | 215 | 234 | | 0.92 | [0.88;0.95] |
| Zhang, W3 | 26 | 137 | | 0.19 | [0.13;0.27] |
| Alshami, AA | 69 | 128 | | 0.54 | [0.45;0.63] |
| Kong, W | 45 | 473 | | 0.10 | [0.07;0.13] |
| Wang, Y2 | 63 | 279 | | 0.23 | [0.18;0.28] |
| **Subgroup estimate** | | **1518** | [0.14;0.51] | **0.29** | **[0.15;0.48]** |
| **Overall estimate** | | **6616** | [0.03;0.67] | **0.20** | **[0.17;0.25]** |

0   0.25  0.5  0.75  1

**Fig 1. Forest plot of proportion ('Prop.') of people with asymptomatic SARS-CoV-2 infection, stratified by setting.** In the setting 'Contact investigations', in which more than one cluster was reported, clusters are annotated with '[cluster]'. The diamond shows the summary estimate and its 95% CI. The red bar and red text show the prediction interval. CI, confidence interval; SARS-CoV-2, severe acute respiratory syndrome coronavirus 2.

hospitalised children, the point estimate was higher (27%, 95% CI 22%–32%, 10 studies) than in adults (11%, 95% CI 6%–19%, 15 studies) (Fig 1). The proportion of asymptomatic SARS-CoV-2 infection estimated in studies of hospitalised patients (35 studies, 19%, 95% CI 14%–25%) was similar to that in all other settings (44 studies, 22%, 95% CI 17%–29%, S5 Fig). To examine publication status, we conducted a sensitivity analysis, omitting studies that were identified as preprints at the time of data extraction (S6 Fig). The estimate of the proportion of asymptomatic infection in all settings (18%, 95% CI 14%–22%) and setting-specific estimates were very similar to the main analysis.

## Contribution of asymptomatic and presymptomatic infection to SARS-CoV-2 to transmission

Five of the studies that conducted detailed contact investigations provided enough data to calculate a secondary attack rate according to the symptom status of the index cases (Fig 3) [36,65,66,90,111]. The summary risk ratio for asymptomatic compared with symptomatic was 0.35 (95% CI 0.1–1.27) and for presymptomatic compared with symptomatic people was 0.63 (95% CI 0.18–2.26) [66,90]. The risk of bias in ascertainment of contacts was judged to be low in all studies.

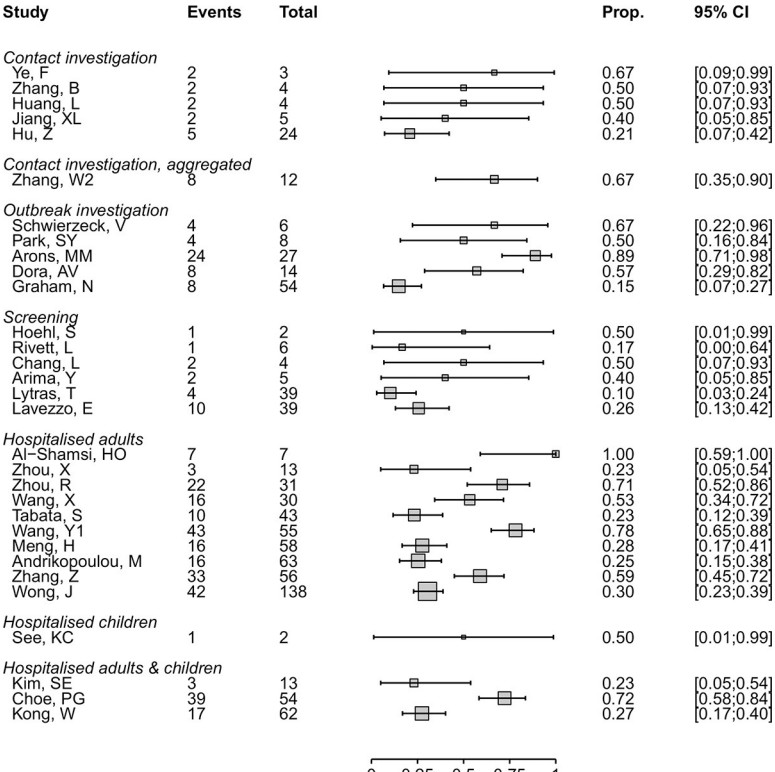

**Fig 2. Forest plot of proportion ('Prop.') of people with presymptomatic SARS-CoV-2 infection, stratified by setting.** CI, confidence interval; SARS-CoV-2, severe acute respiratory syndrome coronavirus 2.

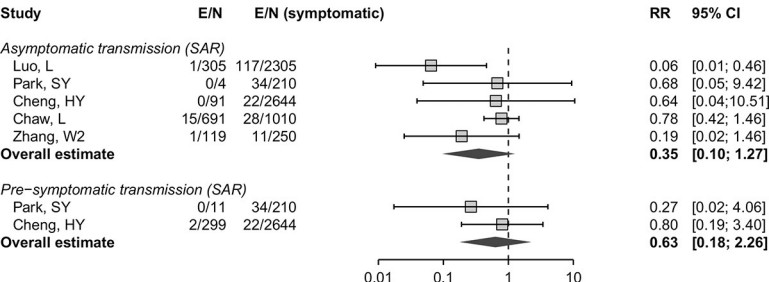

**Fig 3. Forest plot of the RR and 95% CI of the SAR, comparing infections in contacts of asymptomatic and presymptomatic index cases with infections in contacts of symptomatic cases.** The RR is on a logarithmic scale. CI, confidence interval; E, number of secondary transmission events; N, number of close contacts; RR, risk ratio; SAR, secondary attack rate.

We included eight mathematical modelling studies (Fig 4) [19,20,33,51,63,69,78,91]. The models in five studies were informed by analysis of data from contact investigations in China, South Korea, Singapore, and the *Diamond Princess* cruise ship, using data to estimate the serial interval or generation time [19,20,33,69,78], and in three studies the authors used previously published estimates [51,63,91].

Estimates of the contributions of both asymptomatic and presymptomatic infections SARS-CoV-2 transmission were very heterogeneous. In two studies, the contributions to SARS-CoV-2 transmission of asymptomatic infection were estimated to be 6% (95% CrI 0%–57%) [19] and 69% (95% CrI 20%–85%) [69] (Fig 4). The estimates have large uncertainty intervals and the disparate predictions result from differences in the proportion of asymptomatic infections and relative infectiousness of asymptomatic infection. Ferretti and colleagues provide an interactive web application [19] that shows how these parameters affect the model results.

Models of the contribution of presymptomatic transmission used different assumptions about the durations and distributions of infection parameters such as incubation period,

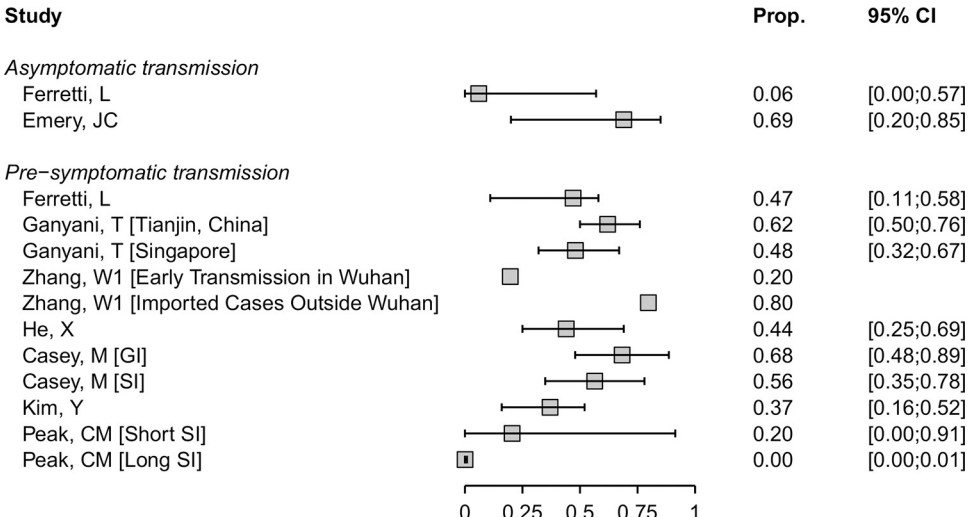

**Fig 4. Forest plot of proportion ('Prop.') of SARS-CoV-2 infection resulting from asymptomatic or presymptomatic transmission.** For studies that report outcomes in multiple settings, these are annotated in brackets. CI, confidence interval; GI, generation interval; SARS-CoV-2, severe acute respiratory syndrome coronavirus 2; SI, serial interval.

generation time, and serial interval [19,20,33,51,63,78,91]. In models that accounted for uncertainty appropriately, most estimates of the proportion of transmission resulting from people with SARS-CoV-2 who are presymptomatic ranged from 20% to 70%. In one study that estimated a contribution of <1% [91], the model-fitted serial interval was longer than observed in empirical studies [115]. The credibility of most modelling studies was limited by the absence of external validation. The data to which the models were fitted were generally from small samples (S7 Fig).

## Discussion

### Summary of main findings

The summary proportion of SARS-CoV-2 that is asymptomatic throughout the course of infection was estimated, across all study settings, to be 20% (95% CI 17%–25%, 79 studies), with a prediction interval of 3%–67%. In studies that identified SARS-CoV-2 infection through screening of defined populations, the proportion of asymptomatic infections was 31% (95% CI 26%–37%, 7 studies). In 31 studies reporting on people who are presymptomatic but who go on to develop symptoms, the results were too heterogeneous to combine. The secondary attack rate from asymptomatic infections may be lower than that from symptomatic infections (relative risk 0.35, 95% CI 0.1–1.27). Modelling studies estimated a wide range of the proportion of all SARS-CoV-2 infections that result from transmission from asymptomatic and presymptomatic individuals.

### Strengths and weaknesses

A strength of this review is that we used clear definitions and separated review questions to distinguish between SARS-CoV-2 infections that remain asymptomatic throughout their course from those that become symptomatic and to separate proportions of people with infection from their contribution to transmission in a population. This living systematic review uses methods to minimise bias whilst increasing the speed of the review process [5,6] and will be updated regularly. We only included studies that provided information about follow-up through the course of infection, which allowed reliable assessment about the proportion of asymptomatic people in different settings. In the statistical synthesis of proportions, we used a method that accounts for the binary nature of the data and avoids the normality approximation (weighted logistic regression).

Limitations of the review are that most included studies were not designed to estimate the proportion of asymptomatic SARS-CoV-2 infection and definitions of asymptomatic status were often incomplete or absent. The risks of bias, particularly those affecting selection of participants, differed between studies and could result in both underestimation and overestimation of the true proportion of asymptomatic infections. Also, we did not consider the possible impact of false negative RT-PCR results, which might be more likely to occur in asymptomatic infections [116] and would underestimate the proportion of asymptomatic infections [117]. The four databases that we searched are not comprehensive, but they cover the majority of publications and we do not believe that we have missed studies that would change our conclusions.

### Comparison with other reviews

We found narrative reviews that reported wide ranges (5%–96%) of infections that might be asymptomatic [1,118]. These reviews presented cross-sectional studies alongside longitudinal studies and did not distinguish between asymptomatic and presymptomatic infection. We found three systematic reviews, which reported similar summary estimates from meta-analysis

of studies published up to May [119–121]. In two reviews, authors applied inclusion criteria to reduce the risks of selection bias, with summary estimates of 11% (95% CI 4%–18%, 6 studies) [120] and 15% (95% CI 12%–18%, 9 studies) [121]. Our review includes all these studies, mostly in the categories of aggregated contact or outbreak investigations, with compatible summary estimates (Fig 1). We categorised one report [81] with other studies in which a defined population was screened. The summary estimate in the third systematic review (16%, 95% CI 10%–23%, 41 studies) [119] was similar to that of other systematic reviews, despite inclusion of studies with no information about follow-up. In comparison with other reviews, rather than restricting inclusion, we give a comprehensive overview of studies with adequate follow-up, with assessment of risks of bias and exploration of heterogeneity (S2–S7 Figs). The three versions of this review to date have shown how types of evidence change over time, from single-family investigations to large screening studies (S1 Table).

## Interpretation

The findings from systematic reviews, including ours [119–121], do not support the claim that a large majority of SARS-CoV-2 infections are asymptomatic [122]. We estimated that, across all study settings, the proportion of SARS-CoV-2 infections that are asymptomatic throughout the course of infection is 20% (95% CI 17%–25%). The wider prediction interval reflects the heterogeneity between studies and indicates that future studies with similar study designs and in similar settings will estimate a proportion of asymptomatic infections from 3% to 67%. Studies that detect SARS-CoV-2 through screening of defined populations irrespective of infection status at enrolment should be less affected by selection biases. In this group of studies, the estimated proportion of asymptomatic infection was 31% (95% CI 26%–37%, prediction interval 24%–38%). This estimate suggests that other studies might have had an overrepresentation of participants diagnosed because of symptoms, but there were also potential selection biases in screening studies that might have overestimated the proportion of asymptomatic infections. Our knowledge to date is based on data collected during the acute phase of an international public health emergency, mostly for other purposes. To estimate the true proportion of asymptomatic SARS-CoV-2 infections, researchers need to design prospective longitudinal studies with clear definitions, methods that minimise selection and measurement biases, and transparent reporting. Serological tests, in combination with virological diagnostic methods, might improve ascertainment of SARS-CoV-2 infection in asymptomatic populations. Prospective documentation of symptom status would be required, and improvements in the performance of serological tests are still needed [123].

Our review adds to information about the relative contributions of asymptomatic and presymptomatic infection to overall SARS-CoV-2 transmission. Since all people infected with SARS-CoV-2 are initially asymptomatic, the proportion that will go on to develop symptoms can be derived by subtraction from the estimated proportion with true asymptomatic infections; from our review, we would estimate this fraction to be 80% (95% CI 75%–83%). Since SARS-CoV-2 can be transmitted a few days before the onset of symptoms [124], presymptomatic transmission likely contributes substantially to overall SARS-CoV-2 epidemics. The analysis of secondary attack rates provides some evidence of lower infectiousness of people with asymptomatic than symptomatic infection (Fig 3) [36,65,66,90,111], but more studies are needed to quantify this association more precisely. If both the proportion and transmissibility of asymptomatic infection are relatively low, people with asymptomatic SARS-CoV-2 infection should account for a smaller proportion of overall transmission than presymptomatic individuals. This is consistent with the findings of the only mathematical modelling study in our review that explored this question [19]. Uncertainties in estimates of the true proportion and

the relative infectiousness of asymptomatic SARS-Cov-2 infection and other infection parameters contributed to heterogeneous predictions about the proportion of presymptomatic transmission [20,33,51,63,78,91].

## Implications and unanswered questions

Integration of evidence from epidemiological, clinical, and laboratory studies will help to clarify the relative infectiousness of asymptomatic SARS-CoV-2. Studies using viral culture as well as RNA detection are needed, since RT-PCR defined viral loads appear to be broadly similar in asymptomatic and symptomatic people [116,125]. Age might play a role as children appear more likely than adults to have an asymptomatic course of infection (Fig 1) [126]; age was poorly reported in studies included in this review (Table 1).

SARS-CoV-2 transmission from people who are either asymptomatic or presymptomatic has implications for prevention. Social distancing measures will need to be sustained at some level because droplet transmission from close contact with people with asymptomatic and presymptomatic infection occurs. Easing of restrictions will, however, only be possible with wide access to testing, contact tracing, and rapid isolation of infected individuals. Quarantine of close contacts is also essential to prevent onward transmission during asymptomatic or presymptomatic periods of those that have become infected. Digital, proximity tracing could supplement classical contact tracing to speed up detection of contacts to interrupt transmission during the presymptomatic phase if shown to be effective [19,127]. The findings of this systematic review of publications early in the pandemic suggests that most SARS-CoV-2 infections are not asymptomatic throughout the course of infection. The contribution of presymptomatic and asymptomatic infections to overall SARS-CoV-2 transmission means that combination prevention measures, with enhanced hand and respiratory hygiene, testing tracing, and isolation strategies and social distancing, will continue to be needed.

## Supporting information

**S1 PRISMA Checklist.**
(DOCX)

**S1 Text. Search strings.**
(DOCX)

**S1 Fig. Flowchart.**
(PDF)

**S2 Fig. Review question 1, forest plot of included studies, by study precision.**
(PDF)

**S3 Fig. Risk of bias in studies included in review question 1 and review question 2.**
(PDF)

**S4 Fig. Review question 2, forest plot of included studies, by study precision.**
(PDF)

**S5 Fig. Review question 1, subgroup analysis comparing studies of hospitalised patients with all other settings.**
(PDF)

**S6 Fig. Review question 1, sensitivity analysis, omitting studies that were preprints at the time of literature search.**
(PDF)

**S7 Fig. Assessment of credibility of mathematical modelling studies.**
(PDF)

**S1 Table. Types of study included in successive versions of the living systematic review, as of 10 June 2020.**
(DOCX)

**S2 Table. Location of studies contributing data to review questions 1 and 2.**
(DOCX)

## Author Contributions

**Conceptualization:** Diana Buitrago-Garcia, Dianne Egli-Gany, Nicola Low.

**Data curation:** Diana Buitrago-Garcia, Dianne Egli-Gany, Michel J. Counotte, Stefanie Hossmann, Hira Imeri, Nicola Low.

**Formal analysis:** Michel J. Counotte, Georgia Salanti.

**Investigation:** Aziz Mert Ipekci.

**Methodology:** Diana Buitrago-Garcia, Dianne Egli-Gany, Michel J. Counotte, Georgia Salanti, Nicola Low.

**Project administration:** Diana Buitrago-Garcia, Dianne Egli-Gany.

**Supervision:** Nicola Low.

**Validation:** Diana Buitrago-Garcia, Dianne Egli-Gany, Michel J. Counotte, Stefanie Hossmann, Hira Imeri, Aziz Mert Ipekci, Nicola Low.

**Writing – original draft:** Diana Buitrago-Garcia, Nicola Low.

**Writing – review & editing:** Diana Buitrago-Garcia, Dianne Egli-Gany, Michel J. Counotte, Stefanie Hossmann, Hira Imeri, Aziz Mert Ipekci, Georgia Salanti, Nicola Low.

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
