## [Editor Report · Decision Letter 0]

12 Jun 2020

Dear Dr Low, 

Thank you for submitting your manuscript entitled "The role of asymptomatic SARS-CoV-2 infections: rapid living systematic review and meta-analysis" for consideration by PLOS Medicine.

Your manuscript has now been evaluated by the PLOS Medicine editorial staff and I am writing to let you know that we would like to send your submission out for external assessment.

Kind regards,

Richard Turner, PhD

Senior editor, PLOS Medicine

rturner@plos.org

---

## [Decision Letter · Decision Letter 1]

24 Jun 2020

Dear Dr. Low,

Thank you very much for submitting your manuscript "The role of asymptomatic SARS-CoV-2 infections: rapid living systematic review and meta-analysis" (PMEDICINE-D-20-02690R1) for consideration at PLOS Medicine. 

Your paper was evaluated by the editors here and sent to independent reviewers, including a statistical reviewer. The reviews are appended at the bottom of this email and any accompanying reviewer attachments can be seen via the link below:

[LINK]

In light of these reviews, we will not be able to accept the manuscript for publication in the journal in its current form, but we would like to invite you to submit a revised version that fully addresses the reviewers' and editors' comments. You will appreciate that we cannot make a decision about publication until we have seen the revised manuscript and your response, and we expect to seek re-review by one or more of the reviewers. 

We hope to receive your revised manuscript by Jul 08 2020 11:59PM. Please email us (plosmedicine@plos.org) if you have any questions or concerns.

Please let me know if you have any questions. Otherwise, we look forward to receiving your revised manuscript soon. 

Sincerely,

Richard Turner, PhD

rturner@plos.org

As one referee suggests, please update the search. 

Please state in some additional detail how you plan to maintain the "living" status of the review. 

Please remove the word "rapid" from the title. We suggest adapting the title to: "Development and transmission of asymptomatic SARS-CoV-2 infections: a living systematic review and meta-analysis".

Please add a new final sentence to the "methods and findings" subsection of your abstract, quoting 2-3 of your study's main limitations. 

We suggest reversing the order of the two sentences making up the "conclusions" subsection of your abstract. 

After the abstract, we will need to ask you to add a new and accessible "author summary" section in non-identical prose. You may find it helpful to consult one or two recent research papers in PLOS Medicine to get a sense of the preferred style. 

You may wish to briefly explain what a "shiny app" is. 

We would be interested to know whether you see any potential issues in including data from preprints. Would it be possible to report sensitivity analyses omitting these data?

Throughout the text, please add p values alongside 95% CI, where available. 

Please adapt reference call outs so that they precede punctuation, e.g. "... a single point [2,3]." (i.e., removing spaces within the square brackets).

Is reference 1 lacking a report number?

Please add the journal details to reference 4. 

Where you list preprints in your reference list, e.g., references 25 and 26, please add "[preprint]".

In the attached PRISMA checklist, please refer to individual items by section (e.g., "Methods") and paragraph number rather than by line or page numbers, as the latter generally change in the event of publication. 

Comments from the reviewers:

*** Reviewer #1: 

Overall comment

This is a timely and well-conducted systematic review addressing an important question. The review follows standard procedures for conduct of a trustworthy review, and the statistical analyses (including subgroups and sensitivity analyses) are appropriate. 

I have two main comments.

First, it is claimed in multiple places that this is a"living review" but this is not a living review according to common definitions (including that used by Elliott et al in Plos Medicine). Rather, it appears that the authors created a database that merges 4 other databases and searched it twice. The review itself is important and credible, and there is no need to claim that it is something that it is not (ie living).

Second, living or not, it would be important to update the search prior to publication. I appreciate that this may require considerable work in terms of data extraction and analysis; however, the last search was conducted in mid April and a number of studies have been published since that time. 

This review is important and deserves to be reported, but the search and analysis should be updated.

Specific comments

Line 35: "using a living evidence database of SARS-CoV-2 literature" What does this mean 

Line 40: Risk of bias was assessed using a questionnaire for modelling studies? Use of a questionnaire to assess bias is unclear? (It is clearer in the text)

 Line 61: Substantial disagreement' is claimed by contrasting quotes from a report in mid February and a news article in April. This is journalism and does not belong in a medical journal.

Line 84: There are multiple places where this review is claimed to be a 'living' review but no definition is given and it is unclear what makes this review 'living' other than the fact that the search has been updated at least once (which is common for most reviews).

Line 95: "We searched the covid-19 living evidence database…"

- If I understand correctly, this is a single database that combines records from 4 databases. To me this doesn't fullfil the definition of a "living evidence database" (the same could otherwise be said just about PubMed), and having searched it twice doesn't make this a 'living systematic review'.

*** Reviewer #2: 

This manuscript provides a valuable contribution to the COVID-19 literature by summarizing in a "living systematic review" the role of asymptomatic infection in SARS-CoV-2 transmission. 

While I have no important concerns about the methods or conclusions, I have a few minor comments. I also wonder if the authors could address in the discussion section a few additional questions that are pertinent:

- Are there relationships among inoculum dose, peak viral load, and likelihood of symptoms? It seems there's a relationship between viral load and infectiousness -- but do we know about this and symptoms? In people or, if no human data, in the macaque infection model?

- Is there evidence to support that the likelihood of symptomatic infection depends on age, comorbidities, or other demographic factors?

Even if we don't have compelling data for these, to what extent might they play a role in the central topics of this manuscript? It might be good to point out in the discussion section types of research beyond the ones reviewed that might inform the manuscript's main topics. 

Minor points:

Abstract

Line 47. Might be worth clarifying that the inference that 40-60% of infections from presymptomatic transmission comes from modeling studies fit to data? 

Line 50. I'm not sure what 'intermediate' means, or what it's intermediate between. Is there a way to restate this quantitatively? 

Discussion.

In the 'implications and unanswered questions' section, it might be worth caveating many of the statements about various interventions that will be needed for controlling transmission. For example, digital contact tracing may not be necessary for control (though in theory it would be helpful) -- it is still unproven in practice. 

*** Reviewer #3: 

I confine my remarks to statistical aspects of this paper. These were well done and I recommend publication. 

Peter Flom

***

[LINK]

---

## [Decision Letter · Decision Letter 2]

29 Jul 2020

Dear Dr. Low,

Thank you very much for re-submitting your manuscript "Asymptomatic SARS-CoV-2 infections: a living systematic review and meta-analysis" (PMEDICINE-D-20-02690R2) for consideration at PLOS Medicine.

I have discussed the paper with editorial colleagues, and it was also seen again by one reviewer. I am pleased to tell you that, provided the remaining editorial and production issues are dealt with, we expect to be able to accept the paper for publication in the journal.

[LINK]

Please let me know if you have any questions. Otherwise, we look forward to receiving the revised manuscript shortly. 

Sincerely,

Richard Turner, PhD

rturner@plos.org

Requests from Editors:

Please let us know if you plan to approach updates of this living systematic review in a similar way to that for your previous paper (https://doi.org/10.1371/journal.pmed.1002611; i.e., via an author-maintained dashboard).

We suggest adding "Occurrence of ..." to your title. Also, amending the title to "... asymptomatic and pre-symptomatic ..." would seem worthwhile.

At line 43, please substitute "non-significantly lower".

At line 53 and 78, we suggest substituting "are symptomatic". 

We ask you to add a sentence in the methods section of your main text to note that ethics approval was not required for this study. 

Please make that "Limitations" at line 346.

Please revisit your reference list. Where appropriate, 6 author names should be listed, followed by "et al". 

Comments from Reviewers:

*** Reviewer #3: 

I recommended publication of the earlier draft and I still do.

Peter Flom

***

[LINK]

---

## [Editor Report · Decision Letter 3]

18 Aug 2020

Dear Prof. Low, 

On behalf of my colleagues and the academic editor, Dr. Nathan Ford, I am delighted to inform you that your manuscript entitled "Occurrence and transmission potential of asymptomatic and pre-symptomatic SARS-CoV-2 infections: a living systematic review and meta-analysis" (PMEDICINE-D-20-02690R3) has been accepted for publication in PLOS Medicine. 

PRODUCTION PROCESS

PRESS

PROFILE INFORMATION

Thank you again for submitting the manuscript to PLOS Medicine. We look forward to publishing it. 

Best wishes, 

Richard Turner, PhD

Senior Editor 

PLOS Medicine

plosmedicine.org